# High precision oxygen isotopes ($\delta^{18}$O) measurements of atmospheric dioxygen using optical-feedback cavity-enhanced absorption spectroscopy (OF-CEAS)

Clément Piel[1], Daniele Romanini[2], Morgane Farradèche[3], Justin Chaillot[3], Clémence Paul[3], Nicolas Bienville[3], Thomas Lauwers[3], Joana Sauze[1], Kevin Jaulin[4], Frédéric Prié[3], Amaëlle Landais[3]

[1] Ecotron Européen de Montpellier (UAR 3248), Centre National de la Recherche Scientifique (CNRS), Université de Montpellier, Campus Baillarguet, Montferrier-sur-Lez, France

[2] Laboratoire Interdisciplinaire de Physique, Univ Grenoble Alpes, CNRS/UGA, Saint-Martin-d'Hères, France

[3] Laboratoire des Sciences du Climat et de l'Environnement, LSCE - IPSL, CEA-CNRS-UVSQ, Université Paris-Saclay, 91191 Gif-sur-Yvette, France

[4] AP2E, 240, rue Louis de Broglie, Les Méridiens Bâtiment A, CS90537, F-13593 Aix-en-Provence, FRANCE

Correspondance : Clément Piel (clement.piel@cnrs.fr)

## **Short summary**

This paper introduces a new optical gas analyzer based on Optical-Feedback Cavity-Enhanced Absorption Spectroscopy technique (OF-CEAS) enabling high temporal resolution and high precision measurement of oxygen isotopes ($\delta^{18}$O) and dioxygen ($O_2$) concentration of atmospheric $O_2$ (respectively 0.06 ‰ and 0.0002 % over 10 minutes integration). The results underscore the good agreement with isotope ratio mass spectrometry measurements and the ability of the instrument to monitor biological processes.

## **Abstract**

Atmospheric dioxygen ($O_2$) concentration and isotopic composition are closely linked to the carbon cycle through anthropic carbon dioxide ($CO_2$) emissions and biological processes such as photosynthesis and respiration. Measurement of isotopic ratio of $O_2$, trapped in ice core bubbles, brings information about past variation in the hydrological cycle at low latitudes, as well as past

productivity. Currently, the interpretation of those variations could be drastically improved with a better (*i.e.* quantitative) knowledge of the oxygen isotopic fractionation that occurs during photosynthesis and respiration processes. This could be achieved, for example, during experiments using closed-biological chambers. In order to estimate the isotopic fractionation coefficient with a good precision, one of the principal limitations is the need for high frequency on-line measurements

of isotopic composition of $O_2$, expressed as $\delta^{18}O$ of $O_2$ ($\delta^{18}O(O_2)$) and $O_2$ concentration. To address this issue, we developed a new instrument, based on the optical-feedback cavity-enhanced absorption spectroscopy (OF-CEAS) technique, enabling high temporal resolution and continuous measurements of $O_2$ concentration as well as $\delta^{18}O(O_2)$, both simultaneously. Minimum of Allan deviation occurred between 10 and 20 minutes while precision reached 0.002 % for $O_2$ concentration and 0.06 ‰ for

$\delta^{18}O(O_2)$, which correspond to the optimal integration time and analytical precision before instrumental drift started degrading the measurements. Instrument accuracy was in good agreement with dual-inlet isotope ratio mass spectrometry (IRMS). Measured values were slightly affected by humidity, and we decided to measure $\delta^{18}O(O_2)$ and $O_2$ concentration after drying the gas. On the other hand, 1 % increase in $O_2$ concentration increased by 0.53 ‰ the $\delta^{18}O(O_2)$. To ensure good quality of

$O_2$ concentration and $\delta^{18}O(O_2)$ measurements we eventually proposed to measure calibration standard every 20 minutes.

## 1.Introduction

Dioxygen ($O_2$) is the second most important constituent of the atmosphere and the evolution of its atmospheric concentration is closely related to the evolution of the carbon cycle through fossil fuel

combustion and biosphere processes (respiration and photosynthesis). The $O_2$ concentration has been shown to decrease by 0.7 % over the last 800 ka probably because of changes of organic carbon burial and oxidation rates with a stabilizing effect of silicate weathering (Stolper et al., 2016). On shorter timescales, the $O_2$ concentration is showing clear seasonal variations anticorrelated with atmospheric carbon dioxide ($CO_2$) concentration, the combination of both $O_2$ and $CO_2$ concentrations enabling to

document the variability of the marine and terrestrial biosphere productivity through calculation of the atmospheric potential oxygen (*e.g.* Goto et al., 2017; Keeling and Manning, 2014; Keeling and Shertz, 1992; Stephens et al., 1998).

The isotopic composition of $O_2$, expressed as $\delta^{18}O$ of $O_2$ ($\delta^{18}O(O_2)$), exhibits variations at the orbital (10,000 years) and millennial scale (Landais et al., 2010; Severinghaus et al., 2009) but no appreciable

variation at the seasonal scale has been evidenced. Past $\delta^{18}O(O_2)$ variability has been linked to

variations in low latitude water cycle and possibly to the variability of the relative proportions of terrestrial and marine productivity (Bender et al., 1994; Extier et al., 2018; Luz and Barkan, 2011). Still, evaluating the relative importance of these two contributions remains difficult since it relies on fractionation factors associated with the biological processes consuming and producing oxygen. These fractionation factors have only been determined for a small number of species and often at the micro-organism scale.

Combining measurements of $\delta^{17}O$ of $O_2$ ($\delta^{17}O(O_2)$) and $\delta^{18}O(O_2)$ permits to have access to gross primary production and is largely used for this purpose in the ocean in combination with the elemental ratio between $O_2$ and argon (Ar), *i.e.* $O_2/Ar$ ratio (Jurikova et al., 2022; Luz and Barkan, 2000; Stanley et al., 2010). In addition, when measuring both $\delta^{17}O(O_2)$ and $\delta^{18}O(O_2)$ in old air trapped in ice cores, we have access to the past variability of the global biosphere productivity although with large uncertainties (Blunier et al., 2002; Brandon et al., 2020; Luz et al., 1999; Yang et al., 2022). Part of this limitation is again linked to our poor knowledge of the isotopic fractionation factors during the biological processes.

In order to progress on the quantitative determination of the oxygen fractionation factors in the biosphere, a good approach is to work on closed biological chambers where plants grow in controlled conditions. This approach was already applied in previous studies, mostly at the micro-organism scale with regular sampling of air from the biological set-up (Guy et al., 1993; Helman et al., 2005; Stolper et al., 2018). At a larger scale such as for a macroscopic study on a terrarium the duration of the experiment needs to be larger to have a quantified signal (Luz et al., 1999).

One of the main limitations when working on the experimental determination of the fractionation factors is that we need numerous measurements of $\delta^{18}O(O_2)$ and $O_2$ concentration to make a precise determination (Paul et al., 2023). However, sampling air from the chamber at high resolution is not convenient since it implies to reduce the quantity of atmospheric air in the biological chamber.

Development of continuous measurements of $O_2$ concentration has permitted to improve the monitoring of $O_2$ concentration in parallel with greenhouse gases. These methods use different techniques such as fuel cell analyzer (Goto et al., 2013) or gas chromatography coupled with a thermal conductivity detector (Tohjima, 2000). More recently, Berhanu et al. (2019) developed an analyzer based on the cavity-ring-down spectroscopy (CRDS) technique which is able to measure the concentration of $O_2$ high precision (standard error of 0.0001 % over 1-minute measurement) on one mode as well as the $\delta^{18}O(O_2)$ with a 0.01 per mil standard error on a second mode. However, $\delta^{18}O(O_2)$

and $O_2$ concentrations are not measured on the same mode and so not at the same time (an instrument software restart is required for switching between modes).

We present here a new instrument based on the optical-feedback cavity-enhanced absorption spectroscopy (OF-CEAS) technique (see Morville et al., 2005), primarily designed and optimized to measure $\delta^{18}O(O_2)$, but also able to measure the atmospheric $O_2$ concentration in parallel. After a presentation of the instrumental design, we discuss the performances of the instrument as well as the influences of humidity on the determination of $O_2$ concentration and isotopic composition and $O_2$ concentration on $\delta^{18}O(O_2)$. Based on our tests, we propose a sequence for calibration of the instrument during series of measurements. This sequence is then used to compare the performance of this instrument to a dual-inlet Isotope Ratio Mass Spectrometry (IRMS) for measurements of $\delta^{18}O(O_2)$ and $O_2$ concentration.

## 2. Material and methods

### 2.1 Instrument description

We present here a new analyzer for $O_2$ exploiting the high sensitivity absorption spectroscopy technique OF-CEAS. This technique is based on a high finesse optical cavity and is analogous to the well-known Cavity-Ring-Down Spectroscopy. It was first described in Morville et al. (2005) (see also Morville et al., 2014) and since then successfully exploited in different laboratories and field applications such as multispecies breath analysis (Ventrillard-Courtillot et al., 2009), analyses of geothermal gases (Kassi et al., 2006), water isotopes measurements in atmospheric air (Lauwers et al., 2024), or analyses of fossil air extracted from ice cores (Faïn et al., 2014). In addition, the simple and compact optical layout of OF-CEAS led to the development of robust devices now commercialized by the AP2E company [www.AP2E.com]. The instrument used in this study is based on an AP2E analyzer customized to satisfy some specific demands, in our case the possibility to measure simultaneously $\delta^{18}O(O_2)$ and $O_2$ concentration because no such analyzer has been commercialized by the company before. To our knowledge this is the first OF-CEAS implementation using a distributed feed-back (DFB) diode laser in the visible range. For the rest, this instrument represents a standard implementation of the OF-CEAS technique as described in above cited publications, therefore we address the interested reader to those references for an OF-CEAS schematic and details of its working principle and operation (in particular Morville et al. 2014, and in addition Lechevallier et al., 2019). In short, the OF-CEAS technique uses a V-shaped high finesse optical cavity which send back to the laser

a fraction of the resonating light. This optical feedback (OF) allows a self-locking of the laser to the successive cavity resonance frequencies during a frequency scan. This enables a less noisy output signal and increased stability compared to what is typically observed with conventional cavity enhanced absorption spectroscopy (CEAS). A simplified schematic of an OF-CEAS V-shaped optical cavity is presented in figure 2B. Variations relative to those publications are the proprietary AP2E electronics and software for system and laser control, acquisition and digitization and analysis of signals. In addition, the AP2E instrument opto-mechanical layout is a revisited version of previous OF-CEAS layouts, of which different versions can be found in the cited works. In any case, the AP2E proprietary implementation follows the previously described OF-CEAS principles of operation, signal analysis, etc., all the way to obtaining absorption spectra like those in Figure 1, which are then fitted by a multi-line spectrum model with polynomial baseline, also previously discussed (Gorrotxategi-Carbajo et al., 2013).

Concerning the most relevant difference relative to previous OF-CEAS implementations, the selected operating wavelength of the DFB diode laser targets a specific region of the $O_2$ absorption band around 760 nm. This so-called oxygen "A" band is the strongest absorption band of this molecule in the visible and infrared spectral range, even though it is still a relatively weak "forbidden" magnetic-dipole transition. In this spectral region, several weak lines of the same transition for the $^{18}O$-$^{16}O$ and $^{17}O$-$^{16}O$ isotopologues are present near-by strong lines of the main $^{16}O$-$^{16}O$ isotopologue. The specific spectral window chosen for our analyzer, which can be covered by a single current-driven laser frequency scan, is displayed in Figure 1, top panel. A second novelty of this instrument, discussed further below, is the presence of a saturated absorption line in the measured absorption spectrum, as visible in Figure 1, bottom panel.

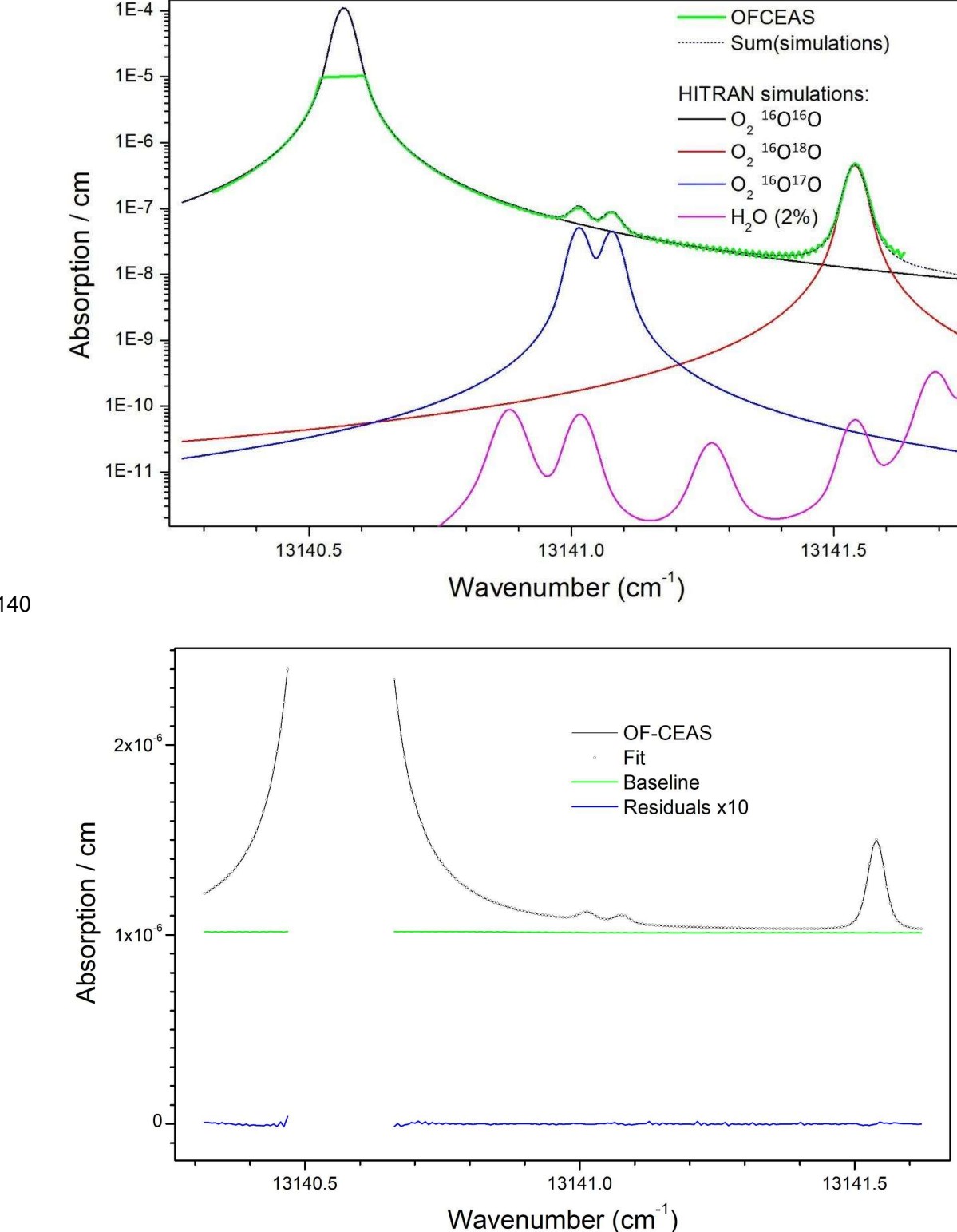


Figure 1: Top: Air absorption spectrum from the instrument (green) superposed with HITRAN2021 simulation (black dotted) which results as the sum of absorption from the main $O_2$ isotopologues and the [18]O and [17]O varieties (respectively in red and blue). Simulated water vapor absorption for a 2 % molar fraction (corresponding to a high level of ambient relative humidity) is plotted to show that it

can be neglected in the spectral analysis. Vertical scale is logarithmic in order to allow clearer display of weaker absorption features. Bottom: OF-CEAS absorption spectrum (black) superposed with spectral fit (black dotted), together with baseline polynomial from fit (green) and fit residuals (data-fit) multiplied by ten (blue).


The analyzer is a modified prototype of a new series of AP2E devices under development at the time of this work, featuring enhanced thermal insulation and stabilization (in particular, sample cell temperature is 37 °C with mK fluctuations). As is the case for most instruments, improved temperature stabilization allows stable measurements on longer time scales, as necessary in this case for averaging

isotopic ratios and detecting their small variations when switching among samples, and in particular for isotopic ratio measurements relative to a reference sample. Since gas sample switching typically implies dead times of about 2 minutes, due to limited flow rates and memory effects, and since at least 5 minutes averaging are additionally required to attain sufficient precision, the shortest time for measurement of one sample is in the range of 5 to 10 minutes. We then need to consider the

instrumental drift (assessed by the Allan deviation presented below) which should remain below the desired precision level over the measurement time for two samples (~ 20 minutes) of which one would be a reference. Smaller drift would afford duty cycle gains by flushing several samples through the analyzer before injection of one calibration sample. The chosen measurement strategy will be discussed later.

The most recent description of an OF-CEAS instrument close to the present one is described in (Lechevallier et al., 2019). The main difference is the visible DFB diode laser emitting at 760 nm (Toptica LD-0760-0040-DFB) in place of the mid-infrared DFB intraband cascade laser (ICL) of that work. Obviously, optical elements are replaced with ones adapted to 760 nm, including a simple room-temperature Si PIN photodiode to monitor cavity transmission. However, no reference photodiode is

installed, with a simulated ramp proficiently replacing the laser power ramp for normalization of cavity transmission signals.

For this setup we obtained from Layertec cavity mirrors (1 m radius of curvature) with a reflection coefficient as good as 0.999956, allowing for ring-down times of up to 30 μs (with room air at the working pressure of 150 mbar, from a 40 cm cavity as in Lechevallier et al. (2019). The sample cell has

a small volume (less than 20 ml) which, combined with a low sample pressure (150 mbar), results in a transit time of about 10 s with a sample flow of only ~15 ml.min$^{-1}$. Another improvement is a more accurate, stable and fast control of the sample pressure inside the measurement cell, which is also important for low drift. In particular, the fast response allows minimization of flow and pressure

perturbations when switching between samples, which contributes to obtaining short commutation
times. Figure 2 shows a picture of the instrument (fig. 2A) and a simplified schematic of the optical
and air flow systems (fig 2B).

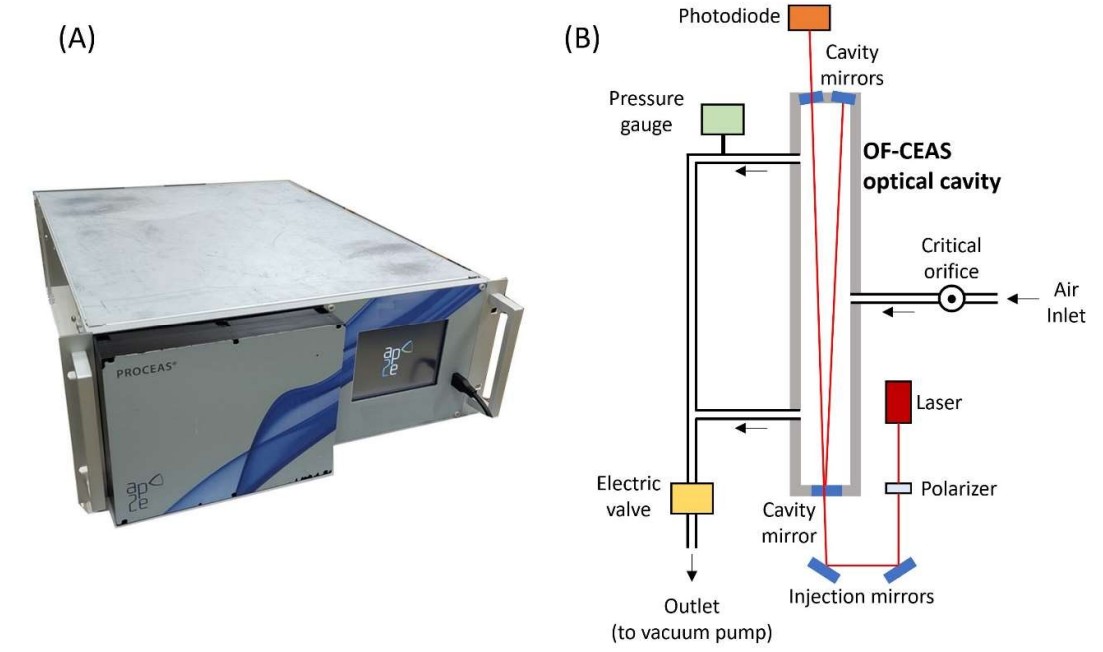

Figure 2: (A) External view of the $O_2$ analyzer: dimensions (without the pump) are 675x485x180mm.
(B) Schematic principle of the analyzer: black arrows are showing the direction of the air stream, and
red lines the laser path in the optical system.

As previously described by Kerstel and Gianfrani (2008) isotopic ratios can be measured by optical
absorption spectroscopy by obtaining spectra containing at the same time the absorption line profiles
of an isotopologue and its main variety. In our case, we are considering the dioxygen isotopologues
mentioned above, in particular the $^{18}O$-$^{16}O$ variety. A simulated absorption spectrum showing in detail
absorption lines for the working spectral region of the instrument is plotted in Figure 1 (based on the
HITRAN spectral database https://hitran.iao.ru/, Gordon et al. 2022) together with a real spectrum
obtained from the instrument. As visible in the bottom panel in Figure 1, the strong $^{16}O$-$^{16}O$ line is
truncated at the top due to saturation of the absorption scale of the instrument (this point will be
addressed below). The isotopic ratio is then obtained as the ratio of the line areas for the considered
isotopologue normalized to that of the main isotopologue. It is straightforward to demonstrate that,
under conditions of constant gas sample temperature and pressure, this ratio is indeed linearly
proportional to the ratio of the mole fractions of these species in the sample. The isotopic ratio

variation (delta value - $\delta$ - in per mil units – ‰) is then calculated after measuring the ratio of areas of the same absorption lines for a reference gas sample relative to which the delta will be defined, by using the Equation 1 (written here for the $^{18}O$-$^{16}O$ case):

$$\delta^{18}O = \left( \frac{\left( \frac{A_{18}}{A_{16}} \right)_{sample}}{\left( \frac{A_{18}}{A_{16}} \right)_{reference}} - 1 \right) \times 1000 \qquad \text{Eq. 1}$$

where $A_{18}$ and $A_{16}$ are the area of the absorption line for $^{18}O$-$^{16}O$ and $^{16}O$-$^{16}O$ respectively.

Given the low abundances of the $^{18}O$-$^{16}O$ (0.2 %) and $^{17}O$-$^{16}O$ (0.04 %) relative to the main isotopologue $^{16}O$-$^{16}O$, the corresponding absorption lines appear much less intense, by about the abundance ratio (Figure 1). This imposes that the instrument sensitivity be high, in order to obtain the absorption lines of these minor isotopologues with sufficient signal to noise ratio. On the other hand, the absorption signal of the main variety will be so strong as to actually exceed the dynamic range of the instrument. Indeed, the high finesse optical cavity containing the sample becomes opaque (no light transmitted) when the laser diode is tuned across the center of any of the $^{16}O$-$^{16}O$ lines available near-by the strongest $^{18}O$-$^{16}O$ and $^{17}O$-$^{16}O$ lines in this absorption band. This problem could be avoided by choosing weaker absorption bands at other wavelengths, but no instrument would be capable as of today of providing a sufficiently good signal for the lines of the minor isotopologues. One way to avoid this would be to use a second diode laser to address a sufficiently weak main isotopologue line lying rather far away in the same absorption band or even in another absorption band (each diode laser covers a small spectral region). However, this would double size and complexity of the experimental setup. Therefore, we decided to rely on the measured wings of the main isotopologue line in order to obtain an estimate of the absorption line area. This proved to be a good strategy given the excellent performance, comparable to those obtained by the OF-CEAS technique with non-saturated absorption lines (see for example isotopic measurements on $H_2O$, Landsberg et al. (2014)). Even if this estimate may be affected by a multiplicative form factor, this same factor will affect measurements of the reference sample and cancel out in the above equation. The situation is particularly favorable for measurement of samples presenting small concentration variations from the target species, as is the case here for dioxygen varying from about 18 to 23 % in biological experiments.

In order to obtain the area of the saturated line of the main isotopologue, we exploit the fact that the data analysis software is able to "count" the cavity modes, each corresponding to a data point in the spectrum (Morville et al., 2014). This works perfectly even in the presence of a large region in the laser spectral scan where points are missing. This is possible thanks to the fact that cavity modes appear uniformly spaced over the laser scan. In addition, we had to take care of the fact that, due to

temperature drift, the cavity length slowly changes. As a consequence, the absolute frequency positions of the cavity modes changes, thus the position of the data points over the absorption line profiles changes too. In particular there are two modes which correspond to the first exploitable data points in the spectrum on the two sides of the strong line, which may go above/below the threshold applied in the analysis software for using them in the spectral fit. Such a change of the number of datapoints in the fit causes a discontinuity in the retrieved line areas used to establish the concentration of the corresponding molecule. In order to fix this problem, we used the position of the data points relative to the absorption lines to continuously adjust (at the mK level) the cavity temperature setpoint and keep the cavity modes at absolute fixed frequency positions (with rms time fluctuations of 1 % of the cavity mode spacings, or about 200 kHz in absolute frequency units). This additionally provides more stable retrieval of all line areas from the spectral fit, as the spectral data points keep fixed positions relative to the absorption profiles which are fitted by Rautian profiles. Even if the Rautian model is much better than the more widely used Voigt profile, it still does not perfectly describe real line profiles. Thus, a displacement of the data points induces small changes in the fit results (even in the absence of noise).

Over the time span of presented results (18 months), we could test two different instrument configurations, but we obtained similar performances. The main difference between the two configurations is the high reflectivity mirrors installed on the V-shaped optical cavity. For the initial configuration and its corresponding set of mirrors (R = 0.999979), the cavity finesse (~74 800) was higher by about a factor 2. The cavity throughput was also higher, which induced a higher level of scattered light and stronger interference fringes on the recorded spectra. The set of mirrors (R = 0.999956), corresponding to the second configuration with lower cavity finesse (F~35 700), had actually also lower transmission (thus relatively higher absorption losses in the coatings). This produced less signal on the photodetector at cavity output but with fractionally less parasitic fringes, partially compensating for the lower finesse. In both cases the setup was carefully optimized, in particular light traps were placed at all positions possibly causing light scatter by secondary reflected beams. In the following, we will specify which setup was used for which results, accounting for somewhat varying performances. It should be noted that for both configurations, the laser scan time over the monitored spectral window displayed in Figure 1 was the same and about 300 ms, while the photodiodes response and digital acquisition time was around 3 μs in order to resolve the profiles of the transmitted cavity modes in the presence of optical feedback, and also the ringdown decay obtained at the end of each laser scan, as detailed in the cited OF-CEAS references.

## 2.2 Air stream selection

The air stream allowed to the instrument inlet was selected using a Valco multiposition valve (EUTF-SD8MWE, VICI AG, Switzerland), ⅛ PFA (Perfluoroalkoxy alkanes) tubing (PFA-T2-030-100, Swagelok), and filtered using a Swagelok 7 μm filter (SS-2F-7). Three gas streams, at ambient pressure, could be selected by the Valco valve: (1) synthetic mixture of $O_2$ and dinitrogen ($N_2$) with an $O_2$ concentration similar to ambient (synthetic dry air, Alphagaz 2, Air Liquide France), (2) ambient atmospheric air (sampled at the inlet of the room ventilation duct) dried with a 40 mL magnesium perchlorate filter, (3) natural dry atmospheric air cylinder enriched in oxygen (Natural Air enriched at 22.9 % $O_2$, Air Liquide Spain) subsequently diluted with $N_2$ (Alphagaz 2, Air Liquide France) using two mass flow controllers (F200CV and F201CV, Bronkhorst, The Netherlands). All the measurements took place in a temperature-controlled room, with temperature fluctuations within +/- 2 °C around setpoint.

When measuring atmospheric air, humid air was dried using a 20 cm long home-made humidity trap made with 6 mm PFA tubing (Swagelok PFA-T6M-1M-30M) and filled by magnesium perchlorate. This trap was daily replaced. For water vapor dependency measurements, the humidity trap was removed, and a stream of synthetic air (Air Product, < 3 ppm of water), with constant $O_2$ concentration and $\delta^{18}O(O_2)$, was humidified at a constant setpoint using a vapor generator (see Leroy-Dos Santos et al., 2021, for more details) up to 9,000 ppm.

## 2.3 Isotope ratio mass spectrometry (IRMS) measurements

IRMS analyses were used as a reference method to validate measurement achieved with OF-CEAS technique. Samples of air were punctually collected in 5 mL glass flasks to be analyzed through IRMS. Operation of the automated sampling system is detailed in Paul et al. (preprint, 2024). Briefly, air from a closed chamber system was circulated through the instrument and the flask for 30 min. Then the flask was automatically isolated thanks to two valves and collected to be analyzed. Results from the flask and the 30 min data acquisition were, in the end, compared.

Flasks were analyzed for $\delta^{18}O(O_2)$ and $\delta O_2/Ar$. $\delta O_2/Ar$ can be converted into $O_2$ concentration (or vice versa) following Equation 2:

$$\delta_{\frac{O_2}{Ar}} = \left( \frac{\left( \frac{[O_2]}{[Ar]} \right)_{sample}}{\left( \frac{[O_2]}{[Ar]} \right)_{reference}} - 1 \right) \times 1000 \qquad \text{Eq. 2}$$

[O$_2$] and [Ar] are the concentrations of O$_2$ and Ar respectively and the reference is the present-day atmospheric air.

Before analysis, samples are purified to isolate O$_2$ and Ar from total air using a gas chromatography column as in Barkan and Luz (2003). Measurements are then performed using a Thermo MAT 253 mass spectrometer (Thermo Scientific) equipped with dual inlet as described in Paul et al. (2023). For each sample, 2 sequences of 16 measurements are performed for measurements of $\delta^{18}O(O_2)$ with a final uncertainty of 0.03 ‰ (1σ). These sequences are followed by two peak jumping sequences to

determine $\delta O_2/Ar$ which is associated with a final uncertainty of 0.5 ‰ (1σ).

## 3. Results and discussion

### 3.1 Precision and drift

The first characterization of the instrument was done through the measurement of the Allan-Werle deviation (Figure 3), which we will call Allan deviation for simplicity. We introduced in the instrument

over 24 hours atmospheric air (dried with a 30 mL magnesium perchlorate trap), which had a constant elemental and isotopic O$_2$ composition. Allan deviation plots were obtained repeatedly for each different configuration, and the result shown is a typical reproducible result. The minimum of the Allan deviation was reached between 10 and 20 minutes of measurements for both O$_2$ concentration and $\delta^{18}O(O_2)$, and with both configurations (Figure 3). The precision achieved for O$_2$ concentration

measurement was better using the initial configuration (0.002 % at 10 minutes averaging; Figure 3 dark green data) than the second configuration (0.005 % at 3 min; Figure 3 light green data). On the contrary, the $\delta^{18}O(O_2)$ minimal Allan deviation was lower with the second configuration than the initial one (0.05 ‰ at 6 min vs 0.08 ‰ at 20 min respectively; Figure 3 grey and dark data).

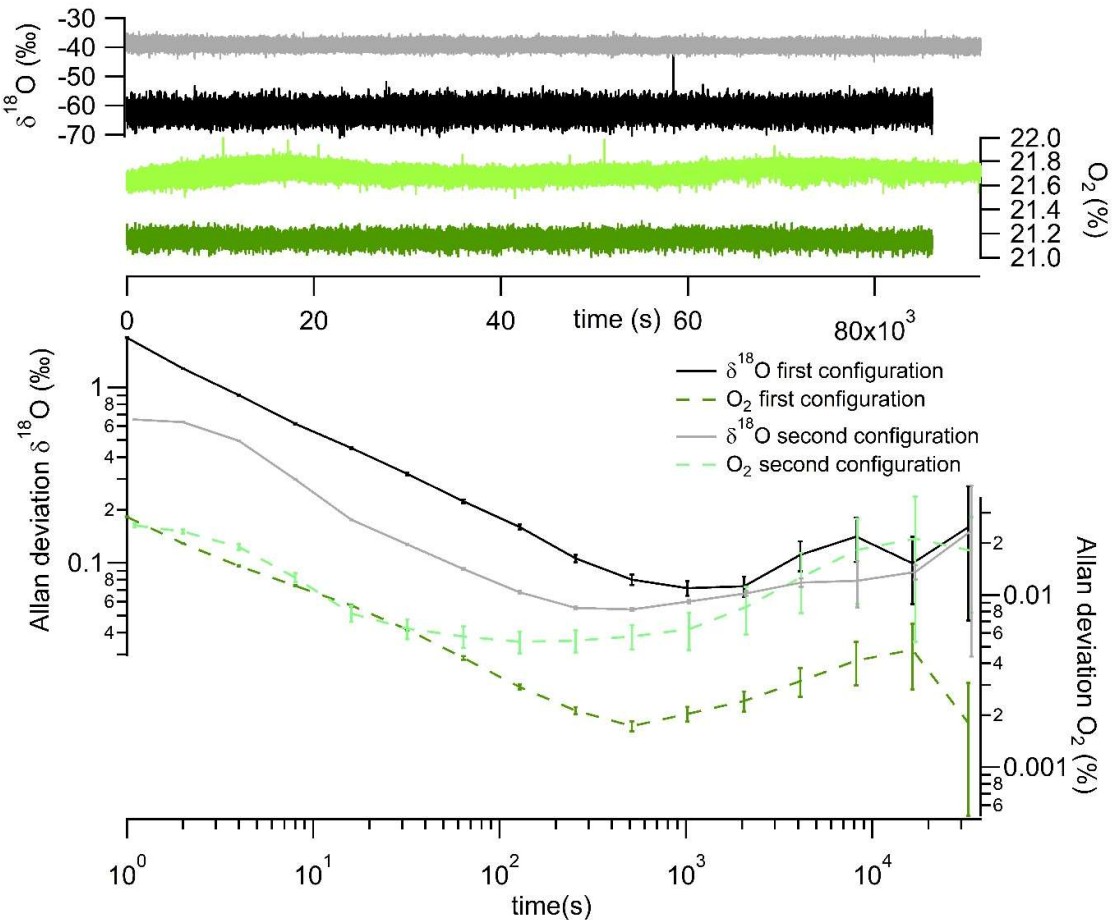

Figure 3: Top panel shows continuous measurement of $\delta^{18}O(O_2)$ (grey and black) and $O_2$ concentration (light and dark green). Bottom panel shows Allan deviation plots based on continuous measurement of both $\delta^{18}O(O_2)$ (grey and black) and $O_2$ concentration (light and dark dotted green). Black and dark green are used for the initial configuration; grey and light green for the second configuration.

After 2 hours, the values of the Allan deviations stayed below 0.2 ‰ for $\delta^{18}O(O_2)$ and 0.03 % for $O_2$ concentration. Considering this drift, in order to maintain measurement repeatability below 0.1 ‰ for $\delta^{18}O(O_2)$ we see that calibrations with a measurement of a standard sample are needed over time intervals of at most 2000 s (30 minutes), in particular for the first configuration (for the second, the Allan deviation for $\delta^{18}O(O_2)$ appears to reach the 0.1 ‰ limit over longer times).

To assess the stability of the instrument on periods longer than 2 hours, we ran measurements of two standards during 8 consecutive days. The 2 standards were dry atmospheric air and a mixing of $O_2$ and $N_2$ in atmospheric proportions (synthetic air from Air Liquide). The evolution of the Allan deviation on Figure 4 was calculated from successive injections of the two standards every 4 minutes. This calibration frequency, higher than required by the Allan deviation discussed above, was chosen as a

compromise towards obtaining measurements with high time resolution. Averages of $\delta^{18}O(O_2)$ and $O_2$ concentration were calculated considering the 2 last minutes (the first two minutes after injection were removed from the calculation to take into account memory effect). With this approach, we obtained an Allan deviation for calibrated measurements of $\delta^{18}O(O_2)$ below 0.03 ‰ after 6 hours of measurements. The evolution of $O_2$ concentration was marked by a diurnal variability which may have

been linked with room temperature changes. The Allan deviation however reached values lower than $1.5 \times 10^{-3}$ % after 15 hours of measurements.

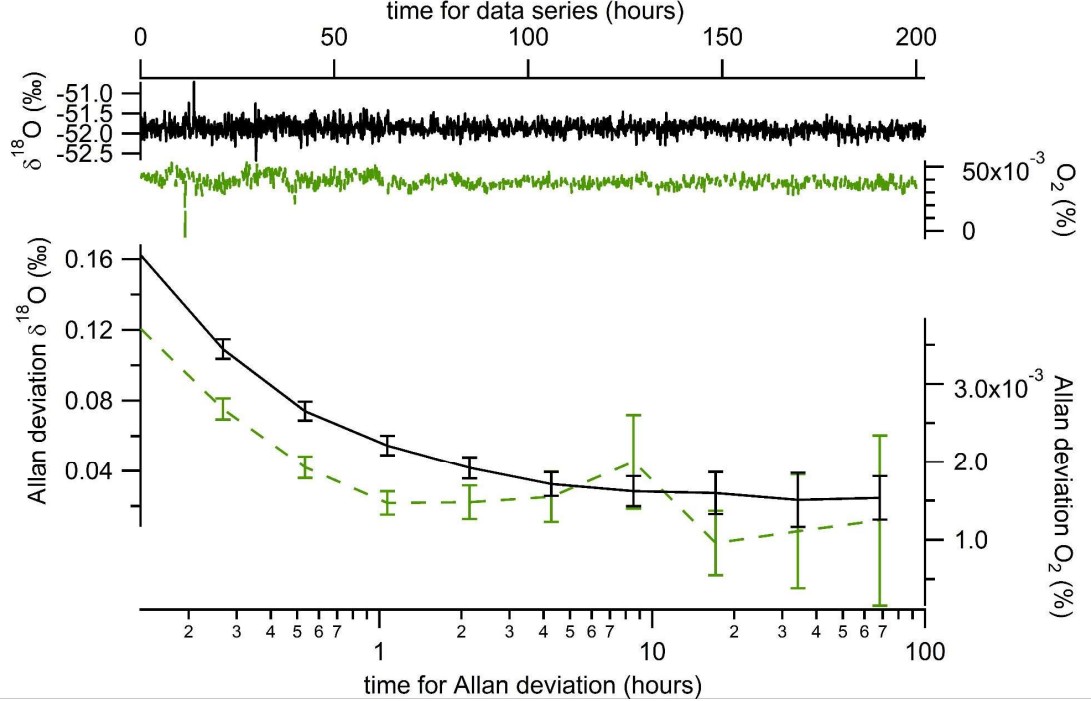

Figure 4: Evolution of difference in mean $\delta^{18}O(O_2)$ (black line) and $O_2$ concentration (green dotted line)

when running successive measurements of two samples (dry atmospheric air and synthetic air) at a periodicity of 8 minutes (4 minutes of atmospheric air followed by 4 minutes of synthetic air). The data used for calculation are the mean values obtained over the last 2 minutes of measurements of each

gas. For this series of measurements, the initial configuration has been used but a similar result was obtained with the second configuration.

3.2 Influence of water vapor concentration

We measured the dependence of $\delta^{18}O(O_2)$ on humidity and found a significant relationship between both (figure 5). Within our range of water vapor mixing ratio (0 to 9 mmol.mol$^{-1}$), a variability of about 1.4 ‰ was found for $\delta^{18}O(O_2)$. The relationship between $\delta^{18}O(O_2)$ and humidity showed a decreasing trend but was highly variable, making any correction unreliable. This dependence of $\delta^{18}O(O_2)$ on

humidity could be expected for two reasons. The first reason is linked to the possible existence of weak water vapor absorption lines. However, there is no water line with sufficient intensity to be significant even at high humidity levels, within the wavelength range analyzed by the instrument. Second, water vapor could affect the linewidth of the oxygen lines by collisional broadening. Indeed, for a 1 % mixing ratio corresponding to high atmospheric humidity, water vapor induces a slightly

enhanced pressure broadening compared to major atmospheric components like $N_2$ or $O_2$. Because it was not easy to propose a correction for the humidity influence, we decided that the most reliable choice for high precision measurements was to measure $\delta^{18}O(O_2)$ after drying the gas flow.

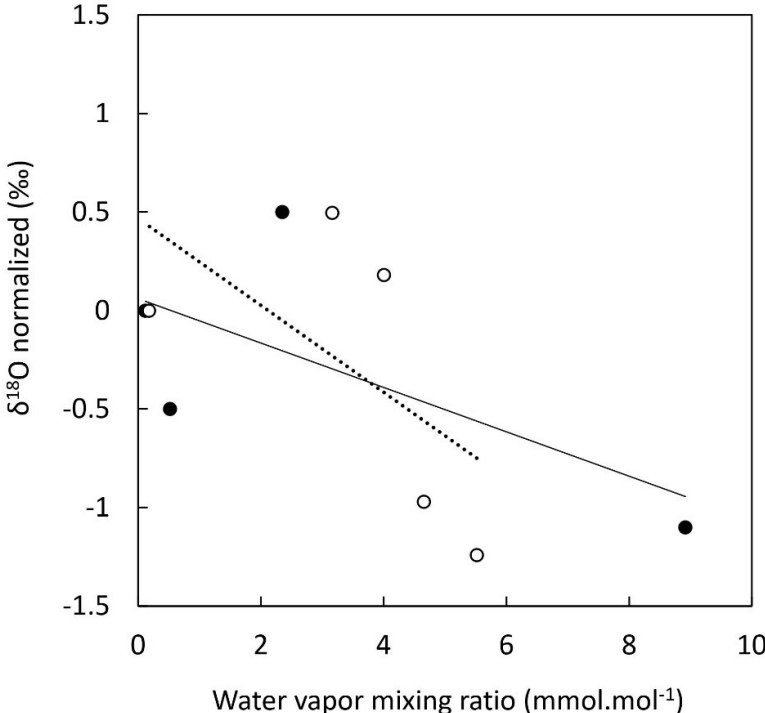

Figure 5: Evolution of $\delta^{18}O(O_2)$ of atmospheric air with increasing water vapor mixing ratio. $\delta^{18}O(O_2)$ was normalized for dry conditions. Two experiments were conducted. Experiment 1: dry atmospheric air was progressively humidified using a vapor generator (filled black dots). Experiment 2: humid atmospheric air was progressively dried using a nafion dryer (hollow dots). Each datapoint is the result of 10min signal integration. Solid line: linear regression for experiment 1 (y=-0.1126x+0.0597, R²=0.45) and dotted line: linear regression for experiment 2 (y=-0.2205x+0.4661, R² = 0.36).

### 3.3 Influence of oxygen concentration

An influence of $O_2$ concentration in the sample on the measured $\delta^{18}O(O_2)$ was expected as is usual in all spectroscopic measurements. Indeed, the parameters in the models used for spectral simulations are associated with uncertainties which do not enable us to reproduce sufficiently well the changes in the shape of the experimental spectrum, in particular the strong saturated $^{16}O$ line as a function of oxygen concentration. In fact, we use a simple Voigt profile which allows us to fit down to the experimental noise level the wings of this strong line which are captured by the measurements. However, rather than trying to figure out the variation of the model parameters as a function of $O_2$ concentration, we found it simpler (and equivalent as a first-order treatment of the problem) to take this effect into account through the study of the influence of $O_2$ concentration directly on the measured $\delta^{18}O(O_2)$, while using constant lineshape parameters. Figure 6 shows the linearity effect, *i.e.* the evolution of $\delta^{18}O(O_2)$ with increasing $O_2$ concentration. Standard dry atmospheric air enriched with $O_2$ was diluted with nitrogen to measure 6 different $O_2$ mixing ratios (Figure 6 x-axis). The difference in $\delta^{18}O(O_2)$ between diluted standard and atmospheric air was then used to correct for any instrument drift (Figure 6 y-axis). The influence of $O_2$ concentration on $\delta^{18}O(O_2)$ was clearly significant. $\delta^{18}O(O_2)$ increased by 0.53 ‰ with 1 % increase in $O_2$ concentration (Figure 6).

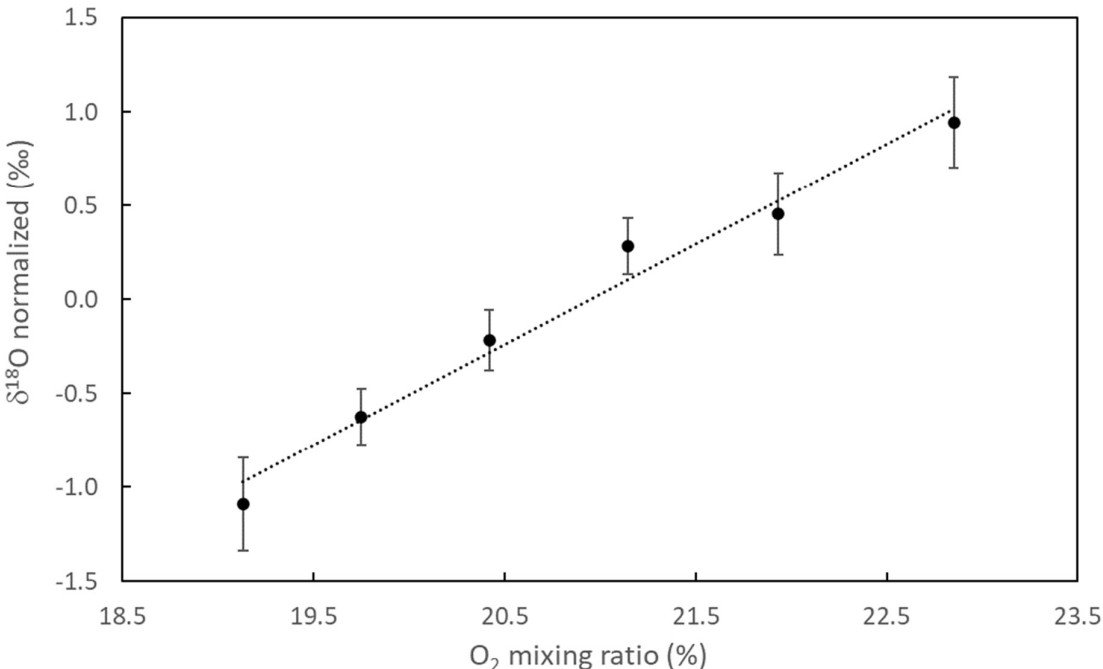

Figure 6: Evolution of the difference in $\delta^{18}O(O_2)$ between a standard (atmospheric air enriched with $O_2$) diluted with $N_2$ and the atmospheric air, vs the $O_2$ concentration of the diluted standard. $\delta^{18}O(O_2)$ is normalized for the ambient $O_2$ mixing ratio, and error bars are the standard deviation of individual measurements (n=27). Typical standard deviation for $O_2$ mixing ratio was 0.01 %. Dotted line: linear regression (y=0.5336x-11.178, $R^2$=0.9782). This result was obtained with the initial configuration (see end of §2.1).

Finally, note that absorption line profiles are influenced by sample parameters such as temperature and pressure. Because of possible long-term drift in pressure and temperature sensors, it is a good practice to regularly check the evolution of this influence of $O_2$ on the measured $\delta^{18}O(O_2)$ (which we did as a precaution once every 15 days).

3.4 Memory effect and response time

The response time of the analyzer and associated inlet system, when switching between different gas streams, was dependent on the total volume of the system (valve, filter, dryer, tubing, and the

instrument internal volume) and the flow rate at the inlet. It was determined that 2 minutes purge time between samples (with a constant flow rate), were necessary to reach a new steady state for both $O_2$ concentration and $\delta^{18}O(O_2)$ (Figure 7).


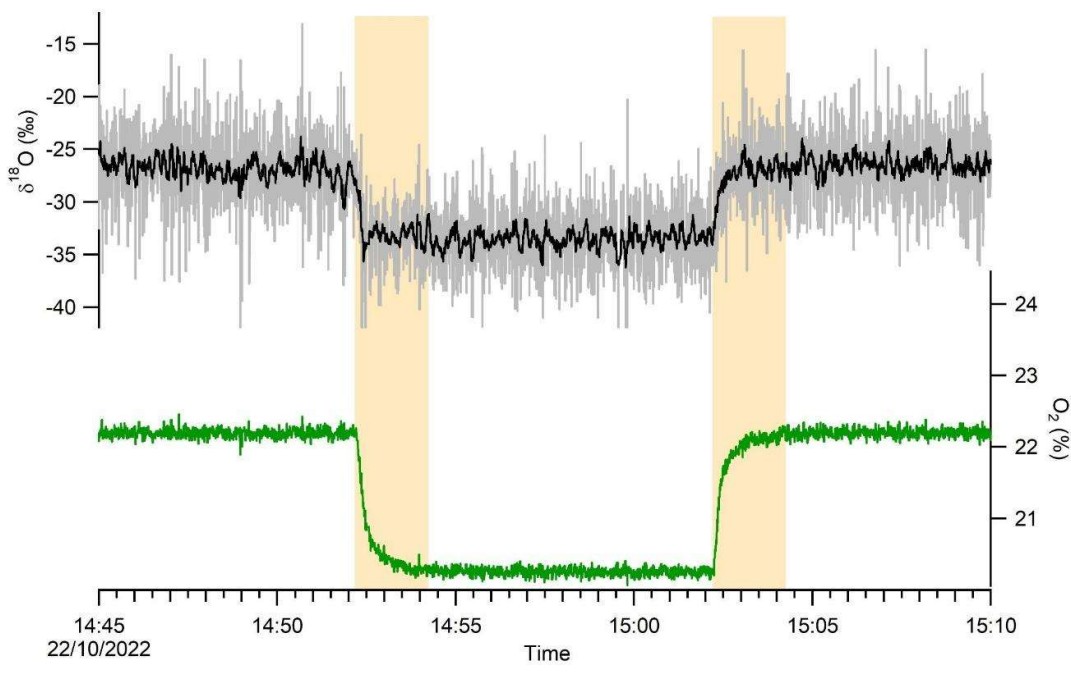

Figure 7: Raw data of $\delta^{18}O(O_2)$ (3 Hz, gray line) and moving average (over 5 second, black line) measured with two gas streams with contrasted $O_2$ concentration (green line). The response time is outlined in yellow. These results were exactly the same for the two configurations.


3.5 Calibration strategy

Based on the detailed results above, we used the following strategy to ensure good quality measurements of the isotopic and elemental composition of $O_2$ in atmospheric air using an OF-CEAS instrument.

First, we always place a magnesium perchlorate trap at the entrance of the instrument to work with dry air, hence easily work with standards made of dry air and avoid the dilution effect observed on

the $O_2$ concentration. Then, for calibration of the signal we need to follow different steps. The dependency of the $\delta^{18}O(O_2)$ on $O_2$ concentration is rather stable and it can be checked regularly by running successive dilution of a standard with $N_2$ (as a precaution it was done every 15 days). On top

of it, a two-point calibration is needed with different values of $\delta^{18}O(O_2)$ and $O_2$ concentration. In our case, we used dry atmospheric air and a standard made of dry atmospheric air enriched with 2 % of $O_2$ (hence $O_2$ concentration of 23 % instead of 21 % in dry atmospheric air). Our tests have shown that the measured difference in $\delta^{18}O(O_2)$ and in $O_2$ concentration between the two standards is stable over a 24 hours period. As a consequence, it is enough to run the two standards only once or twice a day

and then to measure only one standard with the measurements in the daily routine. If the two-point calibration is rather stable over the course of one day, the drift of the instrument occurs on a much shorter timescale. The evolution of the Allan deviation suggested that, to avoid any drift, a one-point calibration should be done at least every 20 minutes. Moreover, to obtain a small $1\sigma$ value (< 0.1 ‰ as mentioned in section 3.1), averages over 5-10 minutes should be done for each sample injection.

Finally, in order to account for the instrument response time, it is important to remove the first 2 minutes from the measurement series (purge time) when we switch from one standard to a sample or from one standard to another standard.

Gathering the recommendations listed above, we suggest using the following measurement sequence:

- Every 15 days: calibration of the influence of $O_2$ concentration on $\delta^{18}O(O_2)$

- Every morning: 6-minutes measurement of atmospheric air enriched with $O_2$ and 6-minutes measurement of atmospheric air. Only the last 4 minutes of each plateau were kept for calculation of the average values.

- During the day: 6-minutes measurements of dry atmospheric air, 6-minutes measurements of
sample 1, 6-minutes measurements of sample 2, 6-minutes measurements of sample 3, back to 6-minutes measurement of dry atmospheric air. Only the last 4 minutes of each plateau were kept for calculation of the average values.

### 3.6 Accuracy of measurements: comparison with IRMS measurements

In order to validate the performance of this new analyzer, with the calibration sequence proposed in the previous section, we measured the same samples with our analyzer and with an IRMS. We sampled air in 5 mL flasks from a closed biological chamber while the analyzer was measuring the elemental and isotopic composition of $O_2$ following the sequence described above. We measured the $\delta^{18}O(O_2)$ from the flasks by IRMS (see section 2.3). These values were compared to the values obtained by the analyzer after corrections of the different influences (cf. previous section) during the same period.

The results of this comparison are displayed on Figure 8 for the initial instrument configuration (same result was obtained for the final configuration). The values obtained for the 3 biological samples by the two techniques align well on a line of slope 1 with zero y-intercept within uncertainty range for measurements of $\delta^{18}O(O_2)$, and with only a slight discrepancy for $O_2$ concentration. This validates the use of the spectroscopy analyzer for ambient air and biological chambers measurements when the values for concentration of $O_2$ do not deviate more than a few % from the current concentration of $O_2$. If the composition of the analyzed air is too different from the current atmospheric air, nonlinear effects on both concentration of $O_2$ and $\delta^{18}O(O_2)$ are expected through modification of the shape of the absorption lines.


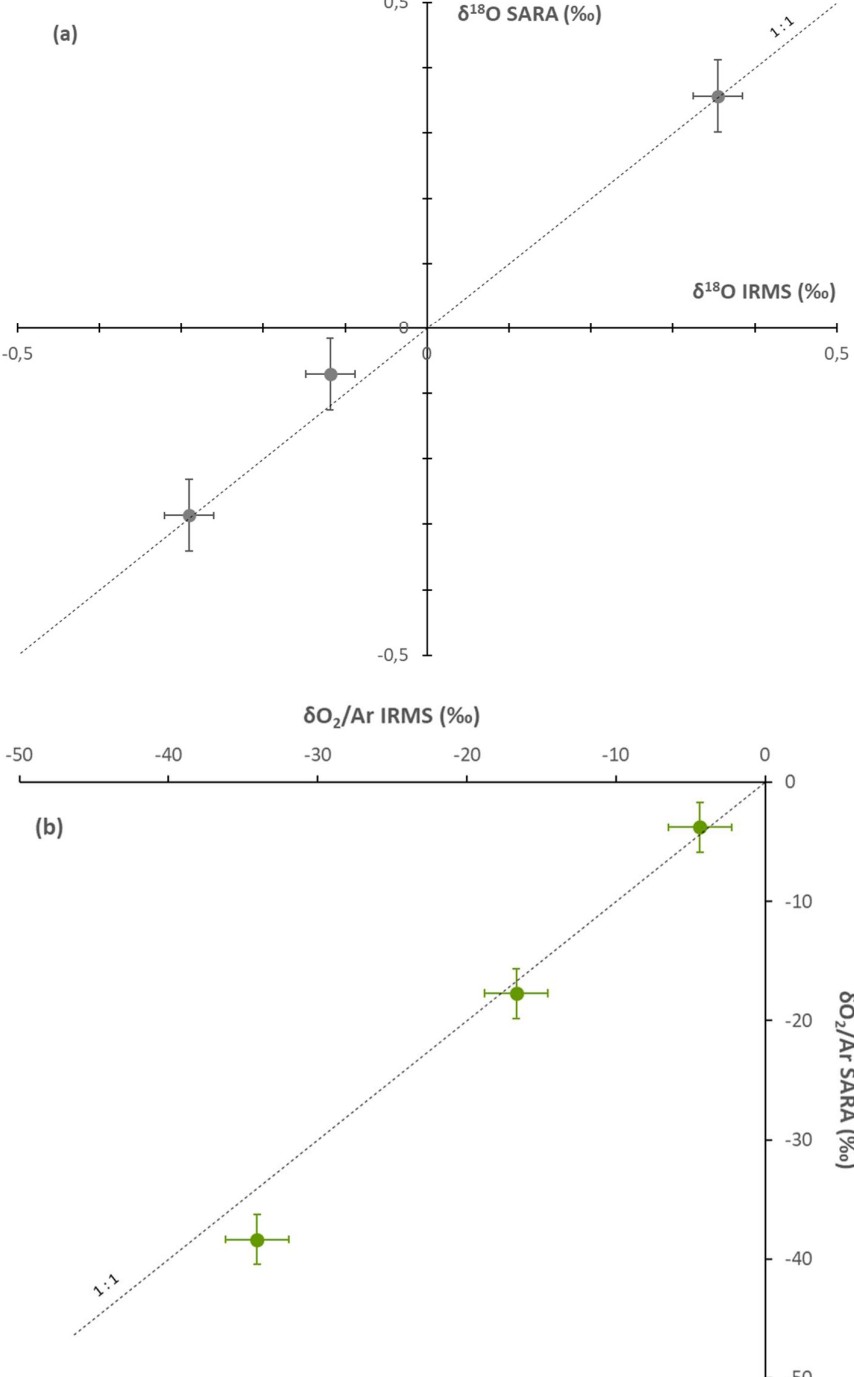

Figure 8: Comparison of the concentration (a) and isotopic composition (b) of $O_2$ obtained by IRMS and by our spectroscopic analyzer (SARA) for different biological samples. The dashed line shows the 1:1 relationship between $\delta^{18}O(O_2)$ measured by IRMS and $\delta^{18}O(O_2)$ measured by the spectroscopic analyzer.

## **4. Conclusion**

We developed a new instrument based on OF-CEAS technique enabling continuous measurements of elemental and isotopic composition of $O_2$ in the atmosphere. We targeted the region of the $O_2$ absorption band around 760 nm to capture a signal for $^{16}O$-$^{17}O$, $^{16}O$-$^{16}O$ and $^{18}O$-$^{18}O$ and developed a special fitting for the $^{16}O$-$^{16}O$ peak which is saturated.

The instrument has been characterized and we could reach a precision of 0.002 % on $O_2$ concentration and 0.05‰ on $\delta^{18}O(O_2)$ for an integration time of 10 minutes. The drift of the instrument deteriorates the signal but the values for Allan deviation stay below 0.2 ‰ for $\delta^{18}O(O_2)$ and 0.03 % for $O_2$ concentration after 2 hours of continuous measurement. $\delta^{18}O(O_2)$ measurements were slightly affected by humidity, and we chose to dry the air flow at the inlet of the instrument. $O_2$ concentration had an influence on $\delta^{18}O(O_2)$ which should hence be regularly quantified.

Based on the performances of the instrument, we proposed a procedure for running $O_2$ measurements for sample or in a continuous way based on a frequent (every 20 minutes) injection of calibration standard. This procedure permitted to obtain performance in good agreement with dual inlet IRMS measurements in a shorter time. It was particularly adapted for monitoring biological processes in a continuous way.

In the future, this instrument may be used in several set-ups where a continuous measurement of $O_2$ concentration and $\delta^{18}O(O_2)$ is needed, such as measurements during biological experiments to follow respiration and photosynthesis evolution, medical set-ups following the evolution of respiration or continuous measurements of the elemental and isotopic composition of fossil air in ice cores. The instrument was also able to measure the $\delta^{17}O(O_2)$ but the current precision was not sufficient yet for competing with IRMS measurements.

## Data availability

The data supporting the conclusions of this paper is available upon request.

## Author contributions

AL and CPi designed the project. DR designed the optical spectrometer, with assistance and expertise from MF and KJ, and most of the hardware was purchased to KJ from AP2E. MF conducted the initial tests and tuning at LiPhy. Final tuning was performed at Liphy by DR and JC. CPi, JS, CPa carried out the experiments at ECOTRON of Montpellier and FP, NB and TL at LSCE. CPi, JS, CPa, NB and AL analyzed the data from the optical spectrometer and CPa and AL analyzed the data from IRMS. CPi, DR, JS and AL prepared the manuscript with contributions from all the authors.

## Competing interests

The authors declare that they have no conflict of interest.

## Acknowledgements

The research leading to these results has received funding from the European Research Council under the European Union H2020 Programme (H2020/20192024)/ERC grant agreement no. 817493 (ERC ICORDA). The authors acknowledge the scientific and technical support of PANOPLY (Plateforme ANalytique géOsciences Paris-sacLaY), Paris-Saclay University, France. This study benefited from the CNRS resources allocated to the French ECOTRONS Research Infrastructure, from the Occitanie Region and FEDER investments as well as from the state allocation 'Investissement d'Avenir' AnaEE- France ANR-11-INBS-0001. This project received fundings from the CNRS Programme National Instrumentation Innovante Transverse (PN IIT) AquaOxy. Nicolas Bienville PhD thesis is funded by the CNRS 80-PRIME interdisciplinarity program FRACOXY between CNRS Ecologie & Environnement (INEE) and CNRS Terre & Univers (INSU). We would also like to thank Sébastien Devidal, Abdelaziz Faez, Olivier Ravel and Alex Milcu from ECOTRON of Montpellier for their help.

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
