# Peer review of "High precision oxygen isotopes ( $\delta^{18}$ O) measurements of atmospheric"

_Atmospheric Measurement Techniques, 2024_

## Referee Comment (RC1)

**General comments**

Piel et al. describe the development of a cavity enhanced spectrometer for the simultaneous measurement of dioxygen concentration and its oxygen-18 isotopic composition. The instrument performances are excellent, reaching detection limits of 0.002 % for $O_2$ concentration and 0.06 ‰ for $\delta^{18}O(O_2)$ in 20 minutes. This development is of significant interest to the scientific community, with multiple environmental applications. However, although the manuscript is highly relevant to AMT's readership and overall well-written, it needs major improvement before being considered for publication.

**Specific comments**

- The instrument description should be rewritten including: a description of the instrument, showing all the elements currently used in the setup (pressure sensor, flow sensor, solenoid valve, mirror, photodiode) and specifying the product reference if commercial. A point-to-point comparison with the reference article should not be made. A technical scheme and an instrument picture should be provided.
- In the results section, there is not enough explanation of how the tests were conducted and there are no data/figures to justify the conclusions given. The link between allan variance and the time used to carry out the measurements is missing. The measurement strategy used should be explained in more detail.
- When results are given, they should be associated with uncertainties and the supplement should explain how they were obtained and the confidence interval chosen.
- Overall, the manuscript is lacking details. It should be revised with additional data to support the development of the instrument.
- The overall structure of the manuscript should be revised. For some sections, the manuscript is written more in the form of a report than a scientific article. The authors should better guide the reader through their instrumental development methodology.
- There are numerous wordings that need to be revised.
- Greater attention should be paid to defining words and acronyms.
- More references are needed throughout the manuscript.

**Technical comments**

All small deltas (δ) must be written in italics as "$\delta$".

**Title**

$\delta^{18}O$ should be defined: "High precision oxygen isotope ($\delta^{18}O$) measurements of …"

**Short summary**

- Line 14: The temporal resolution and precision of measurements should be given.
- Line 15: $\delta^{18}O$ and $O_2$ should be defined.

**Abstract**

- Line 19: "($O_2$)" should be placed after "Atmospheric dioxygen". Then, only $O_2$ should be used throughout the manuscript.
- Line 20: $CO_2$ should be defined.
- Line 24: "isotopic" should be added between "oxygen" and "fractionation". "occur" is missing an "s".
- Line 26: Please add "isotopic" before "fractionation coefficient". Of which isotopic fractionation coefficient are you talking about?
- Line 25: "($\delta^{18}O(O_2)$)" should be added after "$\delta^{18}O$ of $O_2$" and then only $\delta^{18}O(O_2)$ should be used.
- Line 28: Please reverse "OF-CEAS" with "(Optical-Feedback Cavity-Enhanced Absorption Spectroscopy) as "optical-feedback cavity-enhanced absorption spectroscopy (OF-CEAS)". Capital letters are not necessary.
- Line 33: "instrumental" should be added before "drift".
- Line 33-35: need to be more quantitative on humidity and $O_2$ concentration effects.

**Introduction**
- Line 38: "$O_2$" should be added after "Dioxygen" and then only $O_2$ should be used throughout the manuscript.
- Line 48: the $\delta^{18}O$ notation should be defined explicitly. "($\delta^{18}O(O_2)$)" should be added after "$\delta^{18}O$ of $O_2$" and then only $\delta^{18}O(O_2)$ should be used throughout the manuscript. $\delta^{18}O_{atm}$ is useless as it is not used later in the manuscript.
- Line 57-59: $\delta^{17}O$, Ar, and $\Delta^{17}O$ should be defined.
- Line 78: Please reverse "CRDS" with "(Cavity-Ring-Down Spectroscopy) as "cavity ring-down spectroscopy (CRDS)".
- Line 80: As isotopic ratios are expressed in per mill throughout the manuscript, the associated error should be expressed in the same unit.
- Line 83: same comment as for line 28.

**Material and methods**
- Line 95: the reference is not cited correctly, should be "described in Morville et al., (2005)."
- Line 97 : What field applications? References should be given to provide examples
- Line 98 : Simply providing a link to the company is not appropriate. More details should be given.
- Line 99 : "Some specific demands": which ones?
- LIne 100: "DFB" should be defined as "implementation using a distributed feed-back (DFB) diode". Then only DFB should be used.
- Line 108: "the figure" should be replaced with "Figure 1".
- Line 121-124: Data should be provided.
- Line 124 - 126 : "Instrumental drift, assessed by the Allan deviation as presented below, should then remain below the desired precision level over the measurement time for two samples of which one would be a reference". This sentence does not bring necessary information.
- Line 142 : What is "working pressure"? Define here your cavity pressure. "as usual" should be removed.

- Line 144-146 : "Another improvement is a more accurate, stable and fast control of the sample pressure inside the measurement cell, which is also important for low drift." There is an important lack of information here. More elements should be provided. For example, a figure should be added and comparative values for any improvement of the instrument.
- Line 154 : "Well known" should be removed and references for HITRAN spectral database should be given.
- Line 156: "this point will be addressed below". Without going into detail here, a few elements can be given here.
- Line 160: The delta notation should be defined when it is first used in the manuscript, i.e. on line 48. "permil" should be written as "per mill"
- Line 162: "reference sample" of what?
- Line 163: "simple" should be removed and equation numbers should be added
- Line 164: Any equation given in the manuscript should have a number.
- Line 170: How do you obtain the value 35 000 ?
- Line 188-189: what is the software used? "This works well", please be more quantitative.
- Line 198 : What is the frequency dispersion of the cavity modes ? They are not absolutely fixed
- Line 200: References should be provided for the Rautian and Voigt profiles.
- Line 204: "Over the time span of presented results (18 months)". It's not clear what the point of this information is.
- Line 208 : Add the cavity finesse value
- Lien 218-220: The symbol "®" should be added for any deposited trademark cited throughout the manuscript. PFA should be defined. $N_2$ should be defined.
- Line 227: What is the difference between mode (2) line 221 and the routine mode? Why no longer use a trap with magnesium perchlorate?
- Line 232: There is no need for a "-" between "Isotope" and "ratio".
- Line 235: If this manuscript is to be published, the reference given must have been published previously. If this is not the case, further details will be required.
- Line 241: A number should be given to the equation. Besides, the expression to calculate the $O_2$ concentration should be given explicitly.
- Line 248: "is" should be "of". More details are needed for the peak jumping sequences.

**Results and discussion**

- Figure 2 and 3: A different colour palette should be used. Black and green are not colour-blind friendly.
- Line 252-253: This information can be provided earlier and not in the results section.
- Line 254: What is allan deviation? A reference should be provided.
- Line 260-261: The minimum of the allan deviation is not reached at the same time for the oxygen concentration and the isotopy. The time required to reach the minimum for each species must be given with the precision.
- Line 264 : The figure is complicated to understand because of the y-axes
- Line 271: It should be clarified what is considered as a "moderate shift" and "regular measurement.

- Line 276: The time chosen for the measurement must be explained
- Line 277:  How was the time interval between each injection of standard selected?
- Line 284 : The concentration should be kept on the same side of both graphs of figures 2 and 3.
- Line 290: Any results from the secondary configuration should be provided in a supplement.
- Line 293: Data should be provided to support this statement.
- Line 294: This sentence needs rewording.
- Line 300: A reference should be provided. Overall, the structure of section 3.2 should be revised.
- Line 303: The section title should be revised.
- Line 304: This sentence needs rewording.
- Line 314: The linear regression data should be provided in the text. Besides, the given increasing rate of $\delta^{18}O$ with $O_2$ concentration seems wrong based on Figure 4.
- Figure 4: The overall figure display should be improved (e.g., add label ticks, regression equation, …).The symbol for the per mill unity should be used. The errors on the slope and intercept of the linear regression should be provided.
- Line 321: the section number where the initial configuration is described should be added.
- Line 323: Too general, should be more precise.
- LIne 325: Why every 15 days?
- Line 326: The section title should be revised.
- Line 329: The flow rate used for purging must be specified
- Figure 5: there is a typo in the figure legend
- Line 335: Any results from the secondary configuration should be provided in a supplement.
- Line 336: The overall structure of section 3.5 should be revised which is not appropriate for an article.
- Line 353: "small 1σ" should be quantify
- Line 367: This section critically lacks details.

**Conclusion**
- Line 390: The unity used throughout the manuscript should be homogenized.
- Line 400-402: Further details can be given on the instrument's application.

---

## Author Comment (AC1)

**Referee 3**

Many thanks for your detailed review. All our responses or comments are written in green through the text.

This paper reports the development of an optical gas analyzer for high temporal resolution and high precision measurement of δ18O and atmospheric O2 concentration based on the optical-feedback cavity-enhanced absorption spectroscopy (OF-CEAS). The results were compared with the isotope-ratio mass spectrometry (IRMS) and showed good agreement.

**General comments:**

1, Although this work was based on a commercial device (AP2E) and similar experiments have been performed elsewhere, the experimental details still require detailed explanation.

We do give detailed explanations about the special modification of the instrument relative to previous realizations for which we give references where the working principle of the technique and all details about the realization of different prototypes are fully explained. It would be completely redundant to provide an experimental scheme and description of the technique and of the working principle of the instrument which would not be different from what presented in the cited references. In addition, since the presented instrument was realized by a private company we cannot give details concerning the software and the control electronics or the model of pressure and temperature gauges which were selected by the company, which represent proprietary information and some are not available as individual parts (notably, the control electronics). In addition, we note that several works were published in this journal mentioning the exploitation, possibly with modifications, of industrial instruments (chromatographers, mass spectrometers, but also optical devices by Picarro or Aerodyne Research, eg. Berhanu et al., 2019; Kooijmans et al., 2016; Lebegue et al., 2016) without providing detailed descriptions of those instruments.

2, The conclusions of this paper need to be supported by solid experimental data, rather than just stating the conclusions.

The conclusion of the manuscript was improved.

3, A detailed review and comparison with other methods, especially spectroscopic methods, should be made.

We only find one reference of a spectroscopic method for measuring both delta18O and mixing ratio of O2 (see also our answer to the specific comment 2). The other commonly used method is isotope ratio mass spectrometry, also discussed in the manuscript.

**Specific comments:**

1, Page 2, line 32, the argument should contain high accuracy.

We slightly modified the manuscript to explicitly show that our instrument accuracy was compared with calibrated results from dual inlet IRMS.

2, Page 3, for O2 concentration measurement, some other spectroscopy can also achieve ppmv level detection limit, such as Opt. Express 20, 2927, 2012. More discussion and comparison of other high sensitivity laser spectroscopy method is needed.

We refer in this manuscript mainly to the study of Berhanu et al. (2019) using spectroscopy method for measuring O2 concentration in atmospheric air and possibly d18O (O2) but not at the same time than the O2 concentration. We will emphasize more on the performances of this instrument which can be used for similar applications to ours (despite d18O(O2) cannot be measured at the same time as O2 concentration which is a limitation). We have here a lower performance for the O2 concentration since our primary goal was to obtain good performance for the d18O(O2) while still be able to measure O2 concentration at the same time.
We can also cite the work by Brumfield and Wysocki (2012) who developed a faraday rotation spectroscopy instrument for detection of small dioxygen concentration at the level of 1.3 ppmv at integration times of 1 minute. The objective of such a development is different from our development since it aims to detect small amounts of O2 and not variation of O2 concentration (+ d18O(O2)) in atmospheric air.

3, line 80, the difference between two models and parallel measurements is unclear.

Clarified in the text. It refers to the fact that the Picarro instrument operates with 2 modes: O2 high precision (without isotopes) and high precision isotopes (with low precision for O2 mixing ratio). Both modes cannot be operated simultaneously, and an instrument restart is required for switching.

4, line 99, first OF-CEAS in the visible range, incorrect. Salter et al. (Aanlyst, 137, 4669, 2012) performed optical feedback measurement with 635 nm diode laser.

The reference provided describes a cavity enhanced system using optical feedback locking, however the detection is obtained by Raman spectroscopy, thus it is very far from being an OF-CEAS setup where, like here, the detection is made by measuring directly the absorption occurring inside the optical cavity.

5, Page 4, a detailed description of the experimental setup is required.

See answer above.

6, line 136, more discussion about the tuning coefficient and the intensity noise is needed.

We finally decided to remove that paragraph since the development involved in the choice of a diode laser appropriate to the instrument represents excessive details and we believe is of no interest to most readers of AMT. On the other hand, we added the detailed model of the Toptica DFB diode laser that was selected for this instrument, so that the detailed specs of this laser (such as the tuning slope) can be found on the manufacturer's site. We should add that the slope of the laser is not the only critical parameter, thus providing that detail may give the false impression that one needs to satisfy that condition to ensure proper operation of a DFB laser with the OF-CEAS technique.

7, Page 7, page 8, a detailed description of data processing methods is required. The corresponding experimental data need to be presented.

We guess the referee is asking to show examples of raw data obtained in transmission through the V-shaped cavity of the OF-CEAS spectrometer, then display and explain the intermediate steps needed to obtain an absorption spectrum and analyze it. As explained above, all that is already fully explained in cited references about the technique, there would be no new information in adding all those details in the present manuscript. That said, we added in figure 1 a panel with an example of OFCEAS spectrum from the instrument (which was already plotted in the figure together with the HITRAN simulations), together with the spectral fit and the residuals of the fit.

8, line 258, is a data acquisition time of 10 to 20 minutes fast enough for concentration and isotopic measurements?

Our first application for this instrument is the measurements of the evolution of the elemental and isotopic composition of dioxygen in controlled biological chambers. In this kind of experiments, the concentration of O2 and d18O(O2) varies continuously on timescales of several days so that obtaining one accurate data point every 10 or 20 minutes is largely enough. Similarly, this kind of instrument could be used to measure the evolution of O2 concentration and d18O(O2) by continuous flow analysis of gas trapped in ice core (as done for methane by Chappellaz et al., 2013). For this application, it can also be shown that one accurate measurement every 20 minutes is enough to capture the slow temporal variations of O2 concentration and d18O(O2).

9, the influence of water vapor requires experimental data.

Because we remove the water, we only characterize with a few data points the dependency of d18O and O2 on humidity. The graph is provided below and can be included in the revised version of the manuscript if needed. The dependence of the O2 concentration is only due to the dilution of the O2 signal by the added water vapor quantity.

[Figure]

References

Berhanu, T. A., Hoffnagle, J., Rella, C., Kimhak, D., Nyfeler, P., and Leuenberger, M.: High-precision atmospheric oxygen measurement comparisons between a newly built CRDS analyzer and existing measurement techniques, Atmospheric Meas. Tech., 12, 6803–6826, https://doi.org/10.5194/amt-12-6803-2019, 2019.

Kooijmans, L. M. J., Uitslag, N. A. M., Zahniser, M. S., Nelson, D. D., Montzka, S. A., and Chen, H.: Continuous and high-precision atmospheric concentration measurementsof COS, CO$_2$, CO and H$_2$O using a quantum cascade laser spectrometer (QCLS), Atmospheric Meas. Tech., 9, 5293–5314, https://doi.org/10.5194/amt-9-5293-2016, 2016.

Lebegue, B., Schmidt, M., Ramonet, M., Wastine, B., Yver Kwok, C., Laurent, O., Belviso, S., Guemri, A., Philippon, C., Smith, J., and Conil, S.: Comparison of nitrous oxide (N$_2$O) analyzers for high-precision measurements of atmospheric mole fractions, Atmospheric Meas. Tech., 9, 1221–1238, https://doi.org/10.5194/amt-9-1221-2016, 2016.

---

## Author Comment (AC2)

**Referee 1**

Many thanks for your detailed review and your feedback. All our responses or comments are written in green through the text.

**General comments**

Piel et al. describe the development of a cavity enhanced spectrometer for the simultaneous measurement of dioxygen concentration and its oxygen-18 isotopic composition. The instrument performances are excellent, reaching detection limits of 0.002 % for O2 concentration and 0.06 ‰ for δ18O(O2) in 20 minutes. This development is of significant interest to the scientific community, with multiple environmental applications. However, although the manuscript is highly relevant to AMT's readership and overall well-written, it needs major improvement before being considered for publication.

**Specific comments**

● The instrument description should be rewritten including: a description of the instrument, showing all the elements currently used in the setup (pressure sensor, flow sensor, solenoid valve, mirror, photodiode) and specifying the product reference if commercial. A point-to-point comparison with the reference article should not be made. A technical scheme and an instrument picture should be provided.

The manuscript already provides detailed explanations (we added even more details and tried to improve their clarity) about the special modification of the instrument relative to previous realizations, for which we give references where the working principle of the technique and all details about the realization of different prototypes are fully explained. It would be completely redundant to provide an experimental scheme and description of the technique and of the working principle of the instrument which would not be different from what is presented in the cited references. In addition, since the presented instrument was realized by a private company, we cannot give details concerning the software and the control electronics or the model of pressure and temperature gauges which were selected by the company, which represent proprietary information and are not commercially available as individual parts (notably, the control electronics). In addition, we note that several works were published in this journal mentioning the exploitation, possibly with modifications, of industrial instruments (chromatographers, mass spectrometers, but also optical devices by Picarro or Aerodyne Research, eg. Berhanu et al., 2019; Kooijmans et al., 2016; Lebegue et al., 2016) without providing detailed descriptions of those instruments.

● In the results section, there is not enough explanation of how the tests were conducted and there are no data/figures to justify the conclusions given. The link between allan variance and

the time used to carry out the measurements is missing. The measurement strategy used should be explained in more detail.

An effort was made to clarify the result section.

● When results are given, they should be associated with uncertainties and the supplement should explain how they were obtained and the confidence interval chosen.

The Allan variance and response time (fig.5) plots are the results of a single, but highly repeatable measurement. An effort was made to clarify how the results were obtained.

● Overall, the manuscript is lacking details. It should be revised with additional data to support the development of the instrument.

While we added all specific details about this instrument that we could provide, we actually removed mention of the fact that we tested 2 different diode lasers since we realized this point of development was uninteresting to the readers of this journal.

● The overall structure of the manuscript should be revised. For some sections, the manuscript is written more in the form of a report than a scientific article. The authors should better guide the reader through their instrumental development methodology.

We kept the overall structure, but an effort was made to improve the manuscript and better guide the reader.

● There are numerous wordings that need to be revised.

A strong effort was made on this point.

● Greater attention should be paid to defining words and acronyms.

Indeed, we think we fixed all such problems.

● More references are needed throughout the manuscript.

References were added through the manuscript, in particular relative to past applications of the OF-CEAS technique, containing again descriptions of various similar realizations.

**Technical comments**

● All small deltas ($\delta$) must be written in italics as "$\delta$". It has been done in all the manuscript

**Title**

● $\delta18O$ should be defined: "High precision oxygen isotope ($\delta18O$) measurements of …" Done

**Short summary**

● Line 14: The temporal resolution and precision of measurements should be given. Added
● Line 15: δ18O and O2 should be defined. Done

**Abstract**

● Line 19: "(O2)" should be placed after "Atmospheric dioxygen". Then, only O2 should be used throughout the manuscript. Modified in all the manuscript
● Line 20: CO2 should be defined. Done
● Line 24: "isotopic" should be added between "oxygen" and "fractionation". "occur" is missing an "s". Done
● Line 26: Please add "isotopic" before "fractionation coefficient". Of which isotopic fractionation coefficient are you talking about? Done
● Line 25: "(δ18O(O2))" should be added after "δ18O of O2" and then only δ18O(O2) should be used. Modified in all the manuscript
● Line 28: Please reverse "OF-CEAS" with "(Optical-Feedback Cavity-Enhanced Absorption Spectroscopy) as "optical-feedback cavity-enhanced absorption spectroscopy (OF-CEAS)". Capital letters are not necessary. Done
● Line 33: "instrumental" should be added before "drift". Done
● Line 33-35: need to be more quantitative on humidity and O2 concentration effects. We made it quantitative for O2 concentration effect and decided to keep it this way for humidity effect. We explained in more details within section 3.2 the potential effect of water vapor and the solution we applied.

**Introduction**

● Line 38: "O2" should be added after "Dioxygen" and then only O2 should be used throughout the manuscript. Modified in all the manuscript
● Line 48: the δ18O notation should be defined explicitly. "(δ18O(O2))" should be added after "δ18O of O2" and then only δ18O(O2) should be used throughout the manuscript. δ18Oatm is useless as it is not used later in the manuscript. Modified in all the manuscript
● Line 57-59: δ17O, Ar, and Δ17O should be defined. Done
● Line 78: Please reverse "CRDS" with "(Cavity-Ring-Down Spectroscopy) as "cavity ring-down spectroscopy (CRDS)". Done
● Line 80: As isotopic ratios are expressed in per mill throughout the manuscript, the associated error should be expressed in the same unit. Done
● Line 83: same comment as for line 28. Done

**Material and methods**

● Line 95: the reference is not cited correctly, should be "described in Morville et al., (2005)." Done

● Line 97 : What field applications? References should be given to provide examples. Added

● Line 98 : Simply providing a link to the company is not appropriate. More details should be given. We added there a paragraph to clearly explain that the presented instrument actually follows the principle of operation of standard OF-CEAS as described in previous cited works (of which some were added to provide more examples of specific applications as requested above). We hope that now it is clearer that all details, discussions, and schemes of the experimental setup were already provided in previous publications and that it would be completely redundant to add a figure and a couple of pages of explications about OF-CEAS as that are already available in the literature (as much as it is for Mass Spectrometry or, to give a closer case, Cavity Ring-Down spectroscopy for Picarro instruments).

● Line 99 : "Some specific demands": which ones? Now explicitly written in the manuscript

● Line 100: "DFB" should be defined as "implementation using a distributed feed-back (DFB) diode". Then only DFB should be used. Modified in all the manuscript

● Line 108: "the figure" should be replaced with "Figure 1". Done

● Line 121-124: Data should be provided. Done

● Line 124 - 126 : "Instrumental drift, assessed by the Allan deviation as presented below, should then remain below the desired precision level over the measurement time for two samples of which one would be a reference". This sentence does not bring necessary information. We actually think that this sentence is useful since it explains that we need to measure both sample and standard during a period of time not strongly affected by the drift. We have rewritten this sentence to be more precise:

"We then need to consider the instrumental drift (assessed by the Allan deviation as presented below) which should then remain below the desired precision level over the measurement time for two samples (~ 20 minutes) of which one would be a reference. "

● Line 142 : What is "working pressure"? Define here your cavity pressure. "as usual" should be removed. Working pressure is defined on the next line, we added it again after "working pressure". "As usual" refers to OF-CEAS, we changed it to "As usual in OF-CEAS"

● Line 144-146 : "Another improvement is a more accurate, stable and fast control of the sample pressure inside the measurement cell, which is also important for low drift." There is an important lack of information here. More elements should be provided. For example, a figure should be added and comparative values for any improvement of the instrument. This improvement, as specified, was implemented by the AP2E company. Apart from the fact that it is obvious that by improving system parameters stability decreases the measurement drifts in any such kind of spectroscopy-based instrument, we cannot provide additional proprietary information and a comparison of the performance before and after the stability improvement of AP2E instruments. Such tests were conducted at AP2E. We do provide however Allan plots of the presented instrument which illustrate well enough its rather long stability time.

● Line 154 : "Well known" should be removed and references for HITRAN spectral database should be given. HITRAN data base is indeed well-known, so we think we do not need to change the sentence. We add an online reference to the most used HITRAN web site (https://hitran.iao.ru) which is found as a first search result by typing "HITRAN" in Google.

● Line 156: "this point will be addressed below". Without going into detail here, a few elements can be given here. It is "addressed below" because it cannot be addressed in short there. We maintain our text.

● Line 160: The delta notation should be defined when it is first used in the manuscript, i.e. on line 48. "permil" should be written as "per mill". Done

● Line 162: "reference sample" of what? In order to be more precise, we changed it to "...reference gas sample relative to which the delta will be defined, ..."

● Line 163: "simple" should be removed and equation numbers should be added. Modified for all equation of the manuscript

● Line 164: Any equation given in the manuscript should have a number. Modified for all equation of the manuscript

● Line 170: How do you obtain the value 35 000? The cavity finesse is a standard optical cavity parameter which can be obtained from cavity length and measured ringdown time, as described in some of the cited papers. As we actually provide the cavity finesse for the 2 used system configurations in another section of the manuscript following this one, we remove "(35000)" from here as the exact value is actually not important for the discussion at this point.

● Line 188-189: what is the software used? "This works well", please be more quantitative. The software, as mentioned in the added paragraph in a previous paper section, is proprietary of the AP2E instrument. We can't give more details. We changed "well" with "perfectly".

● Line 198 : What is the frequency dispersion of the cavity modes ? They are not absolutely fixed. Actually, as explained, the modes frequencies are determined by the cavity length (as also discussed in cited papers), and since the cavity is temperature stabilized by locking the position of the modes relative to the absorption lines, the modes have very well defined and stable frequencies.

● Line 200: References should be provided for the Rautian and Voigt profiles. References were added.

● Line 204: "Over the time span of presented results (18 months)". It's not clear what the point of this information is. As mentioned, a few lines later: "In the following, we will specify which setup was used for which results, accounting which will account for somewhat varying performances"

● Line 208 : Add the cavity finesse value. For both cavity configurations, finesse and mirror reflectivity are now specified.

● Lien 218-220: The symbol "®" should be added for any deposited trademark cited throughout the manuscript. PFA should be defined. Done

● N2 should be defined. Done

● Line 227: What is the difference between mode (2) line 221 and the routine mode? Why no longer use a trap with magnesium perchlorate? The term "routine mode" was indeed not appropriate. We needed to better precise "when measuring atmospheric air" and in this case, we indeed use magnesium perchlorate filter in the PFA tube. We have corrected it.

● Line 232: There is no need for a "-" between "Isotope" and "ratio". Done

● Line 235: If this manuscript is to be published, the reference given must have been published previously. If this is not the case, further details will be required. It seems to us that sufficient details for the purpose of this paper are given in the rest of the paragraph.

● Line 241: A number should be given to the equation. Besides, the expression to calculate the O2 concentration should be given explicitly. Equation number was added. However, the required expression is trivially derived from the given equation, which is more readable and meaningful than the derived one. We do not see the use of writing out the derived expression.

● Line 248: "is" should be "of". More details are needed for the peak jumping sequences. Thanks! That was corrected.

**Results and discussion**

● Figure 2 and 3: A different color palette should be used. Black and green are not color-blind friendly. We checked using the Coblis Color Blindness Simulator: our figure is visible for all types of color blindness, except monochromacy.

● Line 252-253: This information can be provided earlier and not in the results section. Done, it now also appears in the introduction

● Line 254: What is allan deviation? A reference should be provided. Modified

● Line 260-261: The minimum of the allan deviation is not reached at the same time for the oxygen concentration and the isotopy. The time required to reach the minimum for each species must be given with the precision. Done

● Line 264 : The figure is complicated to understand because of the y-axes. The figure has been modified.

● Line 271: It should be clarified what is considered as a "moderate shift" and "regular measurement. Modified.

● Line 276: The time chosen for the measurement must be explained. A sentence was added: "This calibration frequency, higher than required by the Allan deviation discussed above, was chosen as a compromise towards obtaining measurements with high time resolution."

● Line 277: How was the time interval between each injection of standard selected? See response to previous point.

● Line 284 : The concentration should be kept on the same side of both graphs of figures 2 and 3. Modified.

● Line 290: Any results from the secondary configuration should be provided in a supplement. We do not see any advantage in moving the results of the secondary configuration in a supplement. It seems to us that it takes very little place in the manuscript, and the difference between configuration is clearly stated within the results section.

● Line 293: Data should be provided to support this statement. Because we remove the water, we only characterize with a few data points the dependency of d18O and O2 on humidity. The graph is provided below and can be included in the revised version of the manuscript if needed. The dependence of the O2 concentration is only due to the dilution of the O2 signal by the added water vapor quantity.

[Figure]

● Line 294: This sentence needs rewording. Done

● Line 300: A reference should be provided. Overall, the structure of section 3.2 should be revised. Actually, contrary to what was expected from knowledge on other molecules, a reference was found (and cited) showing that the effect of pressure broadening of O2 lines by water vapor is not much larger than by other atmospheric molecules (O2 itself and N2). Thus, the discussion in this section was modified accordingly.

● Line 303: The section title should be revised. Done

● Line 304: This sentence needs rewording. Done

● Line 314: The linear regression data should be provided in the text. Besides, the given increasing rate of δ18O with O2 concentration seems wrong based on Figure 4. Modified

● Figure 4: The overall figure display should be improved (e.g., add label ticks, regression equation, …). The symbol for the per mill unity should be used. The errors on the slope and intercept of the linear regression should be provided. Done

● Line 321: the section number where the initial configuration is described should be added. Done

● Line 323: Too general, should be more precise. Modified

● Line 325: Why every 15 days?
We did it regularly (i.e. every 15 days) during experiments due to our lack of hindsight on the instrument, which could be considered at that time as a prototype. Clarified in the text.

● Line 326: The section title should be revised. We propose "Memory effect and response time"

● Line 329: The flow rate used for purging must be specified
The flow rate for purging is identical with the one used for measuring. Clarified in the text.

● Figure 5: there is a typo in the figure legend. Corrected

● Line 335: Any results from the secondary configuration should be provided in a supplement.

See our answer for your comment about line 290.

● Line 336: The overall structure of section 3.5 should be revised which is not appropriate for an article. Done.

● Line 353: "small 1σ" should be quantify. Done

● Line 367: This section critically lacks details. We added more details.

**Conclusion**

● Line 390: The unity used throughout the manuscript should be homogenized.
Corrected. We used "‰" everywhere

● Line 400-402: Further details can be given on the instrument's application.
3 examples were given.

References

Berhanu, T. A., Hoffnagle, J., Rella, C., Kimhak, D., Nyfeler, P., and Leuenberger, M.: High-precision atmospheric oxygen measurement comparisons between a newly built CRDS analyzer and existing measurement techniques, Atmospheric Meas. Tech., 12, 6803–6826, https://doi.org/10.5194/amt-12-6803-2019, 2019.

Kooijmans, L. M. J., Uitslag, N. A. M., Zahniser, M. S., Nelson, D. D., Montzka, S. A., and Chen, H.: Continuous and high-precision atmospheric concentration measurementsof COS, CO2, CO and H2O using a quantum cascade laser spectrometer (QCLS), Atmospheric Meas. Tech., 9, 5293–5314, https://doi.org/10.5194/amt-9-5293-2016, 2016.

Lebegue, B., Schmidt, M., Ramonet, M., Wastine, B., Yver Kwok, C., Laurent, O., Belviso, S., Guemri, A., Philippon, C., Smith, J., and Conil, S.: Comparison of nitrous oxide (N2O) analyzers for high-precision measurements of atmospheric mole fractions, Atmospheric Meas. Tech., 9, 1221–1238, https://doi.org/10.5194/amt-9-1221-2016, 2016.

---

## Author Comment (AC3)

**Referee 2**

Many thanks for your feedback. All our responses or comments are written in green through the text.

**General comments**

The article is devoted to measurements of $d^{18}O$ and $O_2$ concentrations in the atmosphere by absorption of DFB laser emission at 760 nm. The excellent sensitivity is presented in short (20minutes) and long (several hours) times. The results of measurements are in nice agreement with that of IRMS. The methods of calibration of the device are suggested for continuous monitoring of $O_2$ concentration and $d^{18}O$.

**Specific comments**

At the same time many details of the experiments are missing.

- Experimental setup is very useful for understanding further.

We refer to previous work for the general instrumental setup, we clarified this point in the manuscript. Thus, no figure or general description of the technique is needed, see our responses to similar comments by the other 2 referees.

- It seems the device long time stability obtained owing to high stability of cuvette (temperature of ~mK fluctuation). Nevertheless, no info in the text about the actual reasons for the stability was provided (precision of measurement, response time of feedback).

We modified the paragraph discussing this point to make it clearer that the whole instrumental optical assembly, including the laser and the pressure gauge, are temperature stabilized, with in particular a cell stability at the mK level. It is well known that by temperature stabilization of an optical setup one reduces the measurement drifts due to changes in optical beam trajectories thus in the signal amplitudes on the photodetectors and also in the amplitude of parasitic interference fringes due to scattered light. This is very general and very well known by developers of optical instruments. Furthermore, and more in general for any measurement scheme, other sample parameters (such as pressure in our case) affect the final measurement. Thus, temperature stabilization also reduces drifts on the measurement (and/or stabilization) of these parameters and thus on the final measurement of concentrations or isotopic ratios. Considering the broad audience of this journal we have added a small section for explaining these considerations.

- Far wings of absorption line contour are determined by Lorentz (apart from pure Doppler) in all models (Voigt, Rautian etc.). The difference in the applied model in the "main body" of line contour ($O^{16}O^{18}$) should be shown. One can estimate using cavity parameters the number of points (cavity modes) on that are not more than 10. (It is possible to see also the fluctuations around $O^{16}O^{18}$ line contour in fig.1.) The question is – is this enough to get the difference in calculation? In text lines 187-203 it is very difficult to understand how it was done. Apart from temperature influence on cavity length mechanical instabilities (for instance by outer pressure fluctuations) should be mentioned.

This comment is not clear to us. In order to make sense of the first part we have to assume the referee is not referring to the $O^{16}O^{18}$ line but to the $O^{16}O^{16}$ line (main isotopologue). Indeed, this is the line having missing points (12 data points actually) due to excessive intensity of the absorption at line center, as explained in the manuscript. As the referee states, the profile is mostly Lorentzian on the line wings which are the only part of this line appearing in the spectrum. The referee then seems to refer to small differences between the HITRAN simulation and the measured spectrum which are visible in the wings of this line, in fig.1. However, the figure does not show the spectral fit but a simple simulation based on spectroscopic line parameters from the literature. The fit of OF-CEAS spectra which are actually used to obtain concentrations and isotopic ratios matches the experimental spectrum much better than the HITRAN simulation and would not be distinguishable from it. To address this point, we added the residuals of the fit as a bottom panel in fig 1.

- Lines 206-215. Why the reflectivity of the second configuration is absent? "Less parasitic fringes" at lower finesse configuration means better spline/averaging of the signal (FSR cavity modes did not change). If it is so, single scan time of the laser frequency and time constant of the detector provided.

Since we specified that the finesse was half we assumed one could easily see what change of mirror reflectivity (R) that corresponds to, given that the finesse is proportional to 1/(1-R). We now explicitly added the reflectivity and cavity finesse for both mirrors sets in this paragraph. However, we are not sure why is the referee speaking about "splines". Anyway, we added a sentence about the laser scan time and the detector response time at the end of the same paragraph.

- Line 304. "An influence of $O_2$ concentration on $d^{18}O$ of $O_2$ was expected." Sound like a general rule, but is only valuable for the method applied.

We are sorry but we do not understand this comment, since the sentences after line 304 explain in detail why we do expect such a dependence for our instrument. If this is general or

not to other methods is not relevant to the discussion. However, we tried to improve this discussion and make it more precise and clearer.

---

## Referee Report (RR1)

The authors provided detailed responses to the review and the manuscript has been significantly improved. However, the revised manuscript requires minor revision before publication. Overall, references to previous work are used excessively. The addition of short descriptive sentences would be appreciated to improve the reading quality of the document. In addition, a thorough proofreading is recommended for typos and wording.

**Abstract**

- Line 30: $\delta^{18}O$ needs to be defined. The short summary, the abstract and the main text are distinct elements where each abbreviation/symbol must be defined independently.

**Intro**

- Line 89-90 : an OFCEAS reference should be placed here
- Line 95: "classical" is not appropriate. IRMS should be defined.

**Material and methods**

A couple of pages are not necessary to give a little more detail on the basic operation of your instrument. On the other hand, and as also notified by the second reviewer, the addition of a schematic diagram, a setup photo and dimensional specifications will certainly help the reader to get a better idea of your new analyzer.

- Line 120 : A reference for spectral fitting is missing here.
- Line 168 : "sccm" must be defined
- Line 173-175 : This sentence needs rewording.
- Line 178 : "well-known" is not needed and the reference to Gordon et al. 2022 should be given.
- Line 191 : "by about the abundance ratio" not exactly, it also depends on the intensity of the transition
- Line 226 : For ease of comparison, the 1 sigma standard deviation of the cavity mode position fluctuation should be indicated. In addition, a graph showing the stability would be valuable for the paper (see Lechevallier et al. 2019).
- Line 295 : ‰, not in ppm
- Line 296 : The bottom subchart in scatter and line could be nice for a clearer view. The label axes should be colored. Why do the uncertainties appear in Figure 3 and not here?
- Line 306 : $\delta^{18}O(O_2)$ should be used instead of "delta".
- Line 315: A multiplication symbol should be used between 1.5 and $10^{-3}$
- Section 3.2 : The data on which this section is based are missing but necessary. Please include a graph with these data in the revised manuscript.
- Line 338 : "was expected as is usual in all spectroscopic measurements". This sentence needs references and rewording.
- Line 346 : Uncertainties on $O_2$ mixing ratios should be indicated.
- Line 415: What is configuration 2?
- Line 426 : Quality parameters for the linear regression should be presented.

---

## Author Response (AR2)

**Thank you very much for your comments, that improved the manuscript. Our answers are in green below.**

The authors provided detailed responses to the review and the manuscript has been significantly improved. However, the revised manuscript requires minor revision before publication. Overall, references to previous work are used excessively. The addition of short descriptive sentences would be appreciated to improve the reading quality of the document. In addition, a thorough proofreading is recommended for typos and wording. **An effort has been made in this direction**

**Abstract**

• Line 30: δ18O needs to be defined. The short summary, the abstract and the main text are distinct elements where each abbreviation/symbol must be defined independently. **Done**

**Intro**

• Line 89-90 : an OFCEAS reference should be placed here **Done**
• Line 95: "classical" is not appropriate. IRMS should be defined. **Done**

**Material and methods**

A couple of pages are not necessary to give a little more detail on the basic operation of your instrument. On the other hand, and as also notified by the second reviewer, the addition of a schematic diagram, a setup photo and dimensional specifications will certainly help the reader to get a better idea of your new analyzer. **A picture and schematic were added (figure 2)**

• Line 120 : A reference for spectral fitting is missing here. **Reference added**
• Line 168 : "sccm" must be defined **Done**
• Line 173-175 : This sentence needs rewording. **Done**
• Line 178 : "well-known" is not needed and the reference to Gordon et al. 2022 should be given. **Done**
• Line 191 : "by about the abundance ratio" not exactly, it also depends on the intensity of the transition. **Indeed since the transition intensities are somewhat different, that is why we say 'by about'. What actually matters are the measured changes to the line intensities which provide linearly the change in the isotopic ratio in the sample. The fact that the intensities are not the same is then irrelevant.**
• Line 226 : For ease of comparison, the 1 sigma standard deviation of the cavity mode position fluctuation should be indicated. In addition, a graph showing the stability would be valuable for the paper (see Lechevallier et al. 2019). **After a closer look, the 1% fluctuations (relative to their spacing) of the cavity mode positions (or optical frequencies) mentioned in the manuscript are just a rough upper limit of the instantaneous fluctuations. The actual standard deviation of the cavity mode position from a laser scan to the next is actually 0.1%, which, multiplied to the cavity mode frequency spacing (187 MHz), gives ~200 kHz rms fluctuations. The manuscript has been modified accordingly. Adding a plot of a flat trace with uniform noise at the 0.1% level is not more informative than the corresponding text statement of this performance which we provide.**
• Line 295 : ‰, not in ppm **Done**

• Line 296 : The bottom subchart in scatter and line could be nice for a clearer view. The label axes should be colored. Why do the uncertainties appear in Figure 3 and not here? **Uncertainties were added to figure 2 as well (now numbered fig 3). Label axes was not colored, but the legend and the axes were improved to make the figure clearer**

• Line 306 : $\delta 18O(O2)$ should be used instead of "delta". **Done**

• Line 315: A multiplication symbol should be used between 1.5 and 10 -3 **Done**

• Section 3.2 : The data on which this section is based are missing but necessary. Please include a graph with these data in the revised manuscript. **A graph was added, and the paragraph 3.2 was modified accordingly**

• Line 338 : "was expected as is usual in all spectroscopic measurements". This sentence needs references and rewording. **In fact there is really not a reference to state this very general fact which is just intrinsic to any physical model of a measured function corresponding to a complex physical phenomenon, as is the case for the spectral absorption profile of a molecule in a thermal gas bath. We reformulated the paragraph in order to make it more evident our approach to the accounting of the change of concentration to the isotopic ratio measurement, which is actually very basic: we just not change the fitting model parameters but we experimentally determine the change of measured ratio of a fixed-ratio sample as a function of its dilution (changing the O2 concentration), which provides a linear slope for corrections in other measurements.**

• Line 346 : Uncertainties on O2 mixing ratios should be indicated. **Added to the legend (uncertainties were about 0.01%, so the error bars are not visible)**

• Line 415: What is configuration 2? **This sentence was reworded**

• Line 426 : Quality parameters for the linear regression should be presented. **This is not a linear regression, but a 1:1 line**